# Changes in Head and Pelvic Movement Symmetry after Diagnostic Anaesthesia: Interactions between Subjective Judgement Categories and Commonly Applied Blocks

**DOI:** 10.3390/ani13243769

**Published:** 2023-12-06

**Authors:** Thilo Pfau, Kaitlyn Sophia Clark, David M. Bolt, Jaclyn Samantha Lai, Melanie Perrier, Jessica Bryce Rhodes, Roger K. Smith, Andrew Fiske-Jackson

**Affiliations:** 1Faculty of Kinesiology, University of Calgary, Calgary, AB T2N 1N4, Canada; 2Faculty of Veterinary Medicine, University of Calgary, Calgary, AB T2N 1N4, Canada; 3Department of Clinical Science and Services, The Royal Veterinary College, Hawkshead Lane, Hatfield AL9 7TA, UK

**Keywords:** horse, lameness, diagnostic anaesthesia, movement symmetry, trot

## Abstract

**Simple Summary:**

Finding the cause of lameness in a horse often involves desensitizing different areas of the limbs, known as diagnostic anaesthesia. When combined with measuring changes in movement, for example, with inertial measurement units, different patterns may emerge in relation to how expert veterinarians perceive these changes. This study has analyzed interactions between perceived changes and measured gait parameters when horses were trotted in-hand in straight lines before and after different types of diagnostic anaesthesia were administered. Our study shows that movement of both the head and pelvis—the most commonly used indicators of lameness—change differently in relation to expert opinion and in association with what part of the limb has undergone elimination of pain. Generally, eliminating pain in a forelimb reduces the asymmetry of head movement (reduced head nodding), but also improves the ability of the horse to utilize the hind limb on the opposite side of the lame forelimb for pushing off from the ground. Eliminating pain in a hind limb appears to improve weight-bearing (supporting the body against gravity) as well as pushing off. This is observed for both the hind and front end of the horse, reducing both hip hike and head nod.

**Abstract:**

Limited evidence is available relating gait changes to diagnostic anaesthesia. We investigated associations between specific movement patterns and diagnostic anaesthesia of different anatomical structures in a retrospective analysis. Referral-level lameness cases were included with the following criteria: presence of diagnostic anaesthesia of a forelimb and/or hind limb; subjective efficacy classified as “negative”, “partially positive”, or “positive”; quantitative gait data available from inertial measurement units. Gait changes were calculated for three forelimb (palmar digital, abaxial sesamoid, low 4-point nerve block) and five hind limb diagnostic blocks (tarso-metatarsal, metatarsophalangeal joint block, deep branch of lateral plantar, low 6-point, abaxial sesamoid nerve block). Mixed models (random factor “case”, fixed factors “diagnostic anaesthesia type” and “efficacy”, two-way interaction) assessed the head and pelvic movement (*p* < 0.05, Bonferroni correction). Four parameters were significantly affected by forelimb anaesthesia (N = 265) (all *p* ≤ 0.031) and six by hind limb anaesthesia (N = 342) efficacy (all *p* ≤ 0.001). All head movement parameters and pelvic push-off asymmetry were significantly affected by the two-way interaction after forelimb anaesthesia (all *p* ≤ 0.023) and two pelvic movement symmetry parameters by the two-way interaction after hind limb anaesthesia (all *p* ≤ 0.020). There are interactions between block efficacy and type resulting in changes in weight-bearing and push-off-associated head and pelvic movement symmetry after diagnostic anaesthesia.

## 1. Introduction

Diagnostic anaesthesia is frequently used in equine lameness investigations with the aim of aiding the identification of anatomical structures or regions involved in a pain-related reduction in locomotor function, a relationship often mentioned as one of the defining criteria of lameness [1]. Visual assessment of the subjective “efficacy” of diagnostic anaesthesia, i.e., the decision on whether or to what degree the use of (regional) anaesthesia has affected the movement of a horse, is affected by bias [2]. Quantitative gait analysis in the form of measuring head and pelvic movement symmetry with body-mounted inertial sensors has provided objective data on changes after diagnostic anaesthesia considered to be negative (no change observed) or positive (some improvement noted) by specialized veterinarians [3]. In this context, the movement symmetry derived from upper-body-mounted inertial sensors has shown strong agreement with the judgement of experienced veterinarians about changes before and after diagnostic anaesthesia [4]. Further differentiation into three categories has been utilized to deal with the outcome encountered in clinical practice: “negative” (no improvement or worsening of the lameness), “partially positive” (some improvement, residual lameness) or “positive” (lameness eliminated or switched to a different limb) [5]. Observers may utilize percentage values to express their judgement about the “efficacy” of diagnostic anaesthesia for altering movement, and a mapping of 0 to 30%, >30 to 70% and >70% improvement has been suggested for the three categories [5].

The use of quantitative movement symmetry analysis before and after diagnostic anaesthesia has generally supported the “law of sides” which provides useful information about the likely source of a lameness: in horses with signs of forelimb and of simultaneous contralateral hind limb lameness, the hind limb lameness is often compensatory, while in horses showing signs of both hind limb and ipsilateral forelimb lameness, the forelimb lameness is often of compensatory nature [6,7]. This information can be useful during clinical lameness examinations, for example, in horses with signs of both forelimb and hind limb lameness, for the decision-making about which limb is a more likely contributor to the displayed lameness pattern. Consequently, when post diagnostic anaesthesia signs of both forelimb and hind limb lameness disappear, the horse has shown signs of a compensatory lameness prior to diagnostic anaesthesia in accordance with the “law of sides”. In this context, it appears interesting to further investigate to what extent a more detailed analysis of quantitative gait collected during clinical lameness examinations may provide further scope for drawing specific conclusions. For example, does the desensitization of different anatomical areas or structures result in different movement patterns of the horse, and how is this reflected in the judgement of veterinary experts?

Furthermore, analysis of clinical lameness cases indicates that the “law of sides” also applies during circular movement (on the lunge) across hard and soft surfaces [8,9]. Linear discriminant analysis suggests that a differentiation between the three “efficacy” categories is supported by circling on a hard surface for forelimb lameness and on a soft surface for hind limb lameness [9]. Interestingly, across different types of anaesthesia, this data-driven approach also shows that data clusters for the three “efficacy” categories show the least amount of separation for straight-line, in-hand trotting. Different responses in overall movement symmetry—i.e., not differentiating between weight-bearing and push-off effects—have also been documented after specific types of distal limb anaesthesia, such as palmar digital +/− abaxial sesamoid nerve blocks vs. distal interphalangeal joint anaesthesia [10]. It hence seems appealing to investigate whether further differentiation between the kinematic changes associated with specific changes in force production (weight-bearing or push-off [11,12]) can help with interpreting straight-line, in-hand trotting data before/after the administration of specific types of diagnostic anaesthesia: does anaesthesia of specific structures or regions create distinct functional changes in weight-bearing and push-off?

The aim of the present study was to conduct a first data-driven retrospective study based on case records investigating the interaction between the functional effects of diagnostic anaesthesia on movement symmetry—quantified using changes in head and pelvic movement symmetry—and the subjective expert judgement of the “efficacy” of the administered anaesthesia, i.e., the overall impression of any improvements noted by the clinicians at the time of the assessment, in conjunction with the diagnostic anaesthesia “type”. We hypothesized that there would be interactions between the diagnostic anaesthesia “type” and subjective judgement of diagnostic anaesthesia “efficacy”—between three categories based on “subjective” expert opinion—in creating functional changes, i.e., in altering the head and pelvic movement symmetry associated with weight-bearing and push-off. We also hypothesized that compensatory movement symmetry changes would—in accordance with the “law of sides”—show contralateral changes after forelimb diagnostic anaesthesia and ipsilateral changes after hind limb diagnostic anaesthesia.

## 2. Materials and Methods

### 2.1. Case Selection

This retrospective study was approved by the Royal Veterinary College Social Sciences Ethical Review Board (URN SR2020-0250). Cases referred to the Royal Veterinary College (RVC) Equine Referral Hospital (ERH) between May 2013 and February 2020 were randomly selected, and after the inspection of case records for (1) the use of inertial sensor based quantitative gait analysis data (EquiGait4w, EquiGait5w or EquiGait8w, EquiGait Ltd., Cheshunt, UK) for straight-line trotting on a hard surface, (2) the (sequential) administration of diagnostic anaesthesia to one (or more) limb(s) and region(s) and (3) the presence of a subjective rating of the diagnostic analgesia, were determined to be eligible for the study. For inclusion into the database, all three conditions had to be met for at least one diagnostic block. For horses with multiple sequential diagnostic blocks, independent of whether they were administered to a single or multiple limbs on the same or different days, all blocks that met all three criteria were included as separate data entries.

### 2.2. Gait Analysis

Symmetry of the head and pelvic vertical movement was quantified using custom written software (EquiGait4w, EquiGait5w or EquiGait8w) implementing published methods [13]. Between four and eight inertial measurement units (Xsens, MTw) were attached to the upper body of each horse using custom-made pouches with double-sided tape (for the thoracolumbosacral area) or Velcro (for the head sensor). The sensor equipment was attached by qualified personnel that had undergone training on sensor attachment. In this study, only data from the four consistently utilized sensors, one attached to the head, one between the tubera sacrale and one each over the left and right tuber coxae, were evaluated. The remaining sensors were attached over the withers, over the thirteenth thoracic vertebra, the third lumbar vertebra and over the caudal sacrum. All data were collected with a minimum sample rate of 60 Hz. Seven movement symmetry measures, three for the head and four for the pelvis, were analyzed. For the head, HDmin, HDmax and HDup quantified the differences in vertical displacement between the minima, maxima or upward movement amplitudes between stride halves; for the pelvis, correspondingly, PDmin, PDmax and PDup were used for the sensor between the tubera sacrale. The fourth pelvic parameter, HHD, quantified the vertical amplitude differences between the left and right tuber coxae sensors.

### 2.3. Data Processing

The sign convention used for the seven movement symmetry measures utilizes negative values for horses with left lameness and positive values for horses with right lameness. For combining data from diagnostic anaesthesia administered to the left and right limbs without the effects of opposite sides canceling out, the following procedure was implemented in accordance with previously published work [8,9]. First, the differences (“Delta” values) between the corresponding symmetry values before and after diagnostic anaesthesia were calculated (Delta = value_after_ − value_before_), rendering negative values for horses with improved symmetry after diagnostic anaesthesia of the right limb, i.e., smaller positive values after administration of the block compared to before. When subsequent blocks were administered to a horse, the values obtained after a specific block were compared to the values obtained before that block rather than to the “baseline” before any blocks had been administered. Second, Delta values of instances where anaesthesia had been administered to the left limbs were inverted (multiplied by negative one). This then also rendered negative the previously positive Delta values for successful blocks being administered to left limbs. The acronyms used for the Delta values for the seven movement symmetry parameters described above are then DHDmin, DHDmax, DHDup, DPDmin, DPDmax, DPDup and DHHD.

As a result of this processing, it is straightforward to draw conclusions about the existence of compensatory changes after successful diagnostic anaesthesia based on the signs of the changes in symmetry in head and pelvic movement. After a successful block to a hind limb, negative values are expected for changes in pelvic movement symmetry in accordance with an improvement in hind limb lameness (or a shift of the lameness to the contralateral hind limb). In accordance with the “law of sides” one would also expect negative values for changes in head movement symmetry reflecting an ipsilateral change. In contrast, after a successful block to a forelimb, negative values are expected for head movement symmetry in accordance with an improvement in forelimb lameness (or a shift of the lameness to the contralateral forelimb). However, now, compensatory changes would be expected to cause contralateral changes in the pelvic movement symmetry, which according to our data processing, would be positive in value.

### 2.4. Subjective Categorisation of Diagnostic Block Effects

Diagnostic blocks were categorized according to their subjectively graded “efficacy” from the case records. Each case contributing to the database was assessed throughout the respective lameness examination by one of four board-certified veterinary surgeons (Dip. ECVS (and ACVS or ECVSMR)).

The case records either directly mentioned the terms “negative”, “partially positive” or “positive” or used percentage improvement values, which were mapped onto the three categories according to [5]: 0 to 30%, >30 to 70% and >70%.

A total of 20 different diagnostic block “types” were administered to at least one forelimb and a total of 16 “types” to at least one hind limb (Figure 1 and Figure 2). For statistical analysis, all block “types” with fewer than 30 instances were categorized as “other”. As a result, three distinct forelimb “types” (palmar digital nerve block (PD), abaxial sesamoid nerve block (ASNB) and low 4-point nerve block (Low4)) plus the category “other” were considered. There were a total of 265 instances of forelimb diagnostic anaesthesia entered into the database (Figure 1), 40 instances for ASNB, 34 instances for Low4, 77 for PD and 114 for “other”. For hind limb anaesthesia, there were five distinct “types” (tarso-metatarsal joint block (TMTJ), deep branch of lateral plantar nerve block (DBLPN), low 6-point nerve block (Low6), metatarsophalangeal joint block (MTPJ) and abaxial sesamoide nerve block (ASNB)), plus the category “other”. There were a total of 342 instances of hind limb diagnostic aneasthesia entered into the database (Figure 2), 30 instances for ASNB, 62 for DBLPN, 58 for Low6, 32 for MTPJ, 102 for TMTJ and 58 for “other”.

### 2.5. Data Analysis

For each occurrence of diagnostic anaesthesia, the case number, the date, the “efficacy” (negative, partially positive, positive), the specific “type” or category “other” (block type), the limb to which it had been administered, as well as the Delta values for the seven movement symmetry parameters were tabulated and termed DHDmin, DHDmax, DHDup, DPDmin, DPDmax, DPDup and DHHD.

Mixed model analysis was conducted using SPSS (v28/29, IBM) with the level of significance set to *p* < 0.05. The case number was used as a random factor, while block “type” and “efficacy” and their two-way interaction were entered as fixed factors. Post hoc analysis was conducted using Bonferroni correction for significant fixed factors and/or their two-way interaction. Estimated marginal means were used to study the size of the changes for “type”, “efficacy” and their two-way interaction.

## 3. Results

A total number of 607 instances of diagnostic anaesthesia (265 to forelimbs, 342 to hind limbs; see Figure 1 and Figure 2 for frequency distribution) from 179 cases was included in the study, ranging per case from 1 instance to a maximum of 16 instances (administered over multiple days and to multiple limbs) (Figure 3). Of the 179 cases, 48 contributed data to the forelimb data set only, 95 contributed data to the hind limb data set only and 36 cases contributed data to both the forelimb and hind limb data sets. Equally, 41 cases contributed data on one forelimb, 84 cases on one hind limb, 8 cases on both forelimbs, 11 cases on both hind limbs, 13 cases on diagonal pairs of limbs, 14 cases on ipsilateral pairs and 8 cases contributed data on more than 2 limbs.

### 3.1. Effect of “Type” and “Efficacy” on Changes in Movement Symmetry after Diagnostic Anaesthesia

#### 3.1.1. Forelimb Diagnostic Anaesthesia

The mixed model analysis shows that all three head movement parameters as well as DPDmax show significant differences as a function of the two-way interaction between forelimb diagnostic anaesthesia “type” and “efficacy” (all *p* <= 0.023) (Table 1). All three indicators of primary forelimb lameness, i.e., parameters related to head movement symmetry, show negative values for partially positive (−2 to −8 mm) and positive blocks (−7 to −19 mm) and marginally positive values for negative blocks (1 to 2 mm). This indicates a reduction in movement asymmetry, i.e., a reduction in lameness in these forelimb lame horses, after partially positive and positive diagnostic anaesthesia. DPDmax, the only pelvic movement parameter significantly affected by diagnostic forelimb anaesthesia, shows marginally positive values for the partially positive and positive “efficacy” categories of on average 1 mm, i.e., a mild reduction in contralateral compensatory hind limb asymmetry in accordance with the “law of sides”.

With respect to diagnostic anaesthesia “type”, the two largest improvements in head movement (highest magnitude negative values) are recorded for low 4-point diagnostic anaesthesia (−13 and −16 mm), followed by ASNB (up to −8 mm) and PD (up to −6 mm) (Table 1).

The combined effects of forelimb diagnostic anaesthesia “type” and “efficacy” are illustrated in Figure 4 for each of the four movement symmetry variables that show significant two-way effects (DHDmin, DHDmax, DHDup, DPDmax, Table 1). The largest negative change in head movement (i.e., a reduction in forelimb lameness) can be seen for positive “efficacy” and Low4 “type” (Figure 4). This condition also shows the highest number of significant pairwise differences in the post hoc analysis of any category and the largest negative pairwise difference (to the negative PD condition) (Appendix A). Positive Low4 diagnostic anaesthesia is also associated with the largest positive compensatory contralateral change for the only pelvic movement symmetry parameter showing a significant two-way effect (DPDmax) (Figure 4, Appendix A).

Negative PD diagnostic anaesthesia appears to create the largest positive change for head movement symmetry. Here, a positive change is indicative of a worsening of lameness (increased lameness grade) attributable to the limb undergoing diagnostic anaesthesia. This condition shows pairwise significant differences to partially positive and positive PD diagnostic anaesthesia, to partially positive and positive Low4 diagnostic anaesthesia and to positive ASNB diagnostic anaesthesia across DHDmin, DHDmax and DHDup (Appendix A).

#### 3.1.2. Hind Limb Diagnostic Anaesthesia

The mixed model analysis identified that two of the four pelvic movement parameters (DPDmin *p* = 0.02, DPDup *p* = 0.019) showed a significant two-way effect of “type” and “efficacy” (Table 2). The other two pelvic movement parameters (DPDmax *p* < 0.001, DHHD *p* = 0.001) as well as two of the head movement parameters (DHDmin *p* < 0.001, DHDup *p* = 0.001) are only significantly affected by “efficacy” but not by diagnostic anaesthesia “type”.

All estimated marginal means (EMM) for the pelvic movement parameters for both partially positive (−3 to −8 mm) and positive diagnostic anaesthesia (−3 to −7 mm) are negative (Table 2) in line with a reduction in hind limb lameness. Similarly, DHDmin and DHDup, the two head movement parameters significantly affected by hind limb diagnostic anaesthesia “efficacy”, show negative EMM values for the partially positive (−5 mm) and positive (−1 to −2 mm) “efficacy” categories, with the negative sign indicative of a small reduction in ipsilateral compensatory head movement asymmetry in accordance with the “law of sides”.

The significant two-way effects of diagnostic anaesthesia “efficacy” and “type” for DPDmin and DPDup are illustrated in Figure 5 and detailed results can be found in Appendix A.

Figure 5 suggests that the most consistent decrease in movement asymmetry from negative to partially positive and positive “efficacy” is appreciable after diagnostic anaesthesia of the deep branch of the lateral plantar nerve (DBLPN) and after administering a low 6-point block. This is apparent for both hind limb weight-bearing (DPDmin) and push-off asymmetry (DPDup). The highest positive asymmetry change values can be observed after negative diagnostic anaesthesia of the metatarsophalangeal joint (MTPJ) (DPDmin and DPDup, Figure 5). With a positive change value indicative of an increase in asymmetry after diagnostic anaesthesia, this indicates a worsening of the lameness attributable to the limb to which diagnostic anaesthesia has been administered. With the exception of DPDmin after an abaxial sesamoid nerve block (ASNB), all other median values for negative “efficacy” appear to be zero or negative (Figure 5).

## 4. Discussion

In the present study, we have quantified changes in head and pelvic movement symmetry with inertial measurement units before and after diagnostic anaesthesia in 179 horses undergoing clinical lameness examinations at a single referral hospital. Previous studies into compensatory patterns during straight-line and circular trotting [4,6,7,8,9,10] have evaluated patterns across a number of block “types” or in relation to the overall movement symmetry changes. The present data-driven, retrospective study aimed at investigating whether different commonly utilized “types” of forelimb and hind limb diagnostic anaesthesia elicit different asymmetry change patterns associated with alterations in weight-bearing and push-off, and as a function of three subjectively expert-assigned “efficacy” categories (negative, partially positive, positive [5]). Other studies [4,6,14] have proposed a value of 50% improvement as the cutoff between negative and positive blocks.

### 4.1. Compensatory Movement Changes after Forelimb Diagnostic Anaesthesia

With respect to compensatory patterns, i.e., changes in pelvic movement after forelimb diagnostic anaesthesia or changes in head movement patterns after hind limb diagnostic anaesthesia, our study confirmed the “law of sides” [6,7]. DPDmax, quantifying movement alterations associated with asymmetrical push-off of the hind quarters [11], was the only pelvic movement parameter significantly affected by diagnostic anaesthesia of a forelimb. The contralateral nature of the pattern, here indicated by a positive Delta value, in conjunction with reducing (or eliminating) forelimb lameness, means that the pelvic limb contralateral to the “blocked” forelimb is able to regain some (or all) of its ability to push off that had been “lost” as a result of the compensatory movement pattern when the horse had shown more prominent lameness before the block. In practice, when using quantitative gait analysis tools, it appears that after a typical “positive” forelimb block, a small reduction in the signs of contralateral hind limb push-off lameness can be observed in the data.

Our findings are in agreement with a previous study investigating forelimb diagnostic anaesthesia, which identified decreases after diagnostic anaesthesia in that parameter of between 66% and 78% for different groups of horses [6]. However, in contrast to the DPDmax push-off effect measured here and previously during straight-line trotting, when quantifying changes after diagnostic anaesthesia of a forelimb across straight-line and circular trotting, three pelvic parameters were affected on the lunge: DPDmin (weight-bearing), DPDup (push-off) and HHD (left-to-right hip amplitude difference) [8]. This is likely related to the different “mechanics” of circular movement, where pelvic push-off asymmetry (termed “MaxDiff” in [15]) shows a considerably shallower slope as a function of the body lean angle, with changes of approximately ±5 mm across a body lean angle range of ±20 degrees, while pelvic weight-bearing asymmetry (termed ‘MinDiff’ in [15]) and hip hike show changes of approximately ±10 to ±15 mm over the same range. While potentially different for forelimb lameness, differences in how individual horses adapt their body lean angle have been shown after hind limb diagnostic anaesthesia [16], which may mask any DPDmax effect when studied across straight-line and circular trotting. When using linear discriminant analysis for studying the effects on movement symmetry of the same three “efficacy” categories used here [9], it is interesting to note that DPDmax gets assigned the highest weight in the first discriminant function for straight-line trot. However, DPDmax is relegated to 7th place (out of 10) overall and to as low as 8th or 9th place for the inside and outside rein on hard ground when measured on the lunge. This confirms that different “mechanics” between straight-line and circular trotting affect pelvic compensatory movement patterns differently. As a result, it appears important to adjust the interpretation of quantitative gait data after diagnostic anaesthesia specifically to whether the horse has been assessed during straight-line or circular movement. The present study would suggest that a change in contralateral compensatory push-off (DPDmax) can be expected after forelimb diagnostic anaesthesia.

### 4.2. Compensatory Movement Changes after Hind Limb Diagnostic Anaesthesia

In analogy to forelimb diagnostic anaesthesia, the present study also confirms the “law of sides” for hind limb diagnostic anaesthesia. In our study, significant effects were measured for DHDmin and DHDup with negative EMM values for partially positive and positive “efficacy” signifying ipsilateral compensation. In practice, when using quantitative gait analysis tools, it appears that after a typical “(partially) positive” hind limb block, a small reduction in the signs of ipsilateral forelimb weight-bearing and push-off lameness can be observed in the data. This indicates that, after a successful block, the horses typically regain (some of) the ability to provide both weight-bearing and push-off forces with the forelimb ipsilateral to the lame hind.

Maliye and co-workers also found evidence of ipsilateral compensation after hind limb diagnostic anaesthesia in horses with hind limb and concurrent ipsilateral forelimb lameness [7]. However, they did not differentiate between weight-bearing- and push-off-related movement patterns (DHDmin, DHDmax, DHDup). In agreement with our results, significant ipsilateral effects were also reported for DHDup on the lunge [8]. However, the latter study also found significant effects for DHDmax, while, in contrast, we report significant changes for weight-bearing-related head movement asymmetry (DHDmin). Linear discriminant analysis has indicated that compensatory head movement in response to hind limb diagnostic anaesthesia is particularly important for lunge movement with the lame limb on the inside of the circle (or quantified as an average across both reins) [9]. This again emphasizes the potential of altered “mechanics” during circular movement as a diagnostic aid for veterinarians and, in analogy to forelimb diagnostic anaesthesia, interpretation of quantitative gait data after diagnostic anaesthesia should be specific to the whether the horse is exercised in straight lines or on the lunge.

### 4.3. Effect of Forelimb Diagnostic Anaesthesia “Type”

Four movement symmetry variables, three associated with head movement and one associated with pelvic movement, showed a significant two-way interaction between diagnostic anaesthesia “type” and “efficacy” (see Table 1).

Visually (see Figure 4), the change in movement symmetry after the low 4-point block appears most different from the remaining “types” showing the highest negative changes for head movement and positive compensatory changes for pelvic movement (DPDmax) in association with positive “efficacy”. This is apparent across all three head movement parameters, suggesting changes in both weight-bearing (DHDmin) and push-off (DHDmax, DHDup). The two largest negative changes across forelimb conditions and all parameters are seen for low 4-point diagnostic anaesthesia for DHDmin with −13.4 mm and DHDup with −16.1 mm (Table 1). Again, this may suggest alterations in weight-bearing and push-off after this “type” of diagnostic anaesthesia.

It also seems noteworthy that for low 4-point diagnostic anaesthesia, visually (Figure 4), there is a clear difference between the positive category and the two other “efficacy” categories. The positive low 4-point diagnostic anaesthesia also has the highest number of pairwise significant differences (eight for DHDmin, seven for DHDmax and six for DHDup, Appendix A) of any “efficacy” and “type” two-way combination. For the other “types” (ASNB and PD), there appears to be a more gradual decrease from the negative to the partially positive and the positive “efficacy” category.

The other forelimb condition that qualitatively and visually shows a somewhat unique pattern is PD diagnostic anaesthesia, with the most positive values for the negative category for all three head movement symmetry differences (DHDmin, DHDmax, DHDup) (Figure 4). This indicates that, for a PD block to be judged as “negative”, horses typically show signs of a “worsening” lameness, although the change is on average of rather small magnitude. This is in agreement with the measured symmetry changes reported previously after PD or ASNB forelimb diagnostic anaesthesia [3].

### 4.4. Effect of Hind Limb Diagnostic Anaesthesia Type

Two pelvic movement symmetry variables (DPDmin, DPDup) revealed a significant two-way interaction between “type” and “efficacy” and one pelvic movement symmetry parameter (DPDmax) showed a one-way effect of “type” (Table 2). Generally, there appeared to be more significant two-way pairwise differences for DPDup, a kinematic parameter associated with push-off, than for DPDmin, a kinematic parameter associated with weight-bearing (Appendix A). The visually most consistent changes as a function of “efficacy”, i.e., increasingly negative Delta values from “negative” to “partially positive” and “positive”, were observed for DBLPN and low 6-point diagnostic anaesthesia.

Similar to the changes after negative forelimb PD diagnostic anaesthesia, positive movement asymmetry changes were apparent after a negative MTP joint hind limb diagnostic anaesthesia. Again, this indicates that, after administering MTP joint anaesthesia with the primary cause of the lameness likely elsewhere in that limb, lameness may worsen (Figure 5).

After ASNB hind limb anaesthesia, a somewhat unexpected pattern arose with larger negative changes (“improvements”) in association with partially positive “efficacy” and smaller-magnitude negative changes (or even positive changes) after positive “efficacy” (Figure 5). This warrants further investigation as it may be related to the temporal sequence of different types of diagnostic anaesthesia. This has not been the focus of the present study andthe same effect was not observed after forelimb ASNB (Figure 4).

### 4.5. Magnitude of Changes

In general, the measured changes in movement symmetry after diagnostic anaesthesia for the different “efficacy” categories and block “types” reported in Table 1 and Table 2 are of small to moderate magnitude with absolute change values up to 18.5 mm for head movement and up to 7.5 mm for pelvic movement, however with generally large variation (Figure 4 and Figure 5). While these maximal values exceed suggested lameness thresholds [17], the majority of reported average asymmetry changes are considerably smaller also in comparison to between-measurement variation in non-lame horses [18] and in a mixed group of horses [19]. Among other reasons, the ‘uncertainty’ for assigning lameness grades for mildly lame horses [20] as well as temporal effects of diagnostic anaesthesia [10] may also contribute to the variability around the average values reported here. Another factor that has likely influenced the magnitude of changes is the initial lameness grade for the first administered block or the residual lameness grade after subsequent blocks. Including the lameness grade–at each stage of the lameness examination–into the statistical analysis for example as a covariate would appear to be useful to investigate this aspect. Furthermore, future studies should investigate the association of gait changes in groups of horses with specific musculoskeletal deficits. In clinical practice, smaller changes in pelvic movement symmetry values should be expected after hind limb diagnostic anaesthesia compared to changes in head movement asymmetry after forelimb anaesthesia.

### 4.6. Limitations

This investigation has utilised three previously defined diagnostic anaesthesia “efficacy” categories based on expert veterinary judgement [5,8,9]. Of particular relevance here is, that each veterinary specialist in charge of a specific case had access to the results of the quantitative gait analysis system (EquiGait4w, EquiGait5w or EquiGait8w) during the lameness workup. It can hence not be excluded that the quantitative results have influenced the decision-making process about the “efficacy” judgement. In this context, it is important to note that the gait analysis results are typically not available immediately due to the workflow of the specific gait analysis system. Depending on the number of sensors used and the length of the exercise analyzed, data processing will take approximately 30 s before being displayed. This provides the clinician with some time to establish and document their opinion about the “efficacy” of the block. Further, when producing comparison charts between gait results before and after diagnostic anaesthesia, movement asymmetry values are presented in ‘real-world’ millimeters in this gait analysis system, rather than percentage change values or normalized movement symmetry values in relation to the range of motion [21]. How much mental processing time is needed for an experienced observer to form an opinion and how much it may be influenced by quantitative gait data and thus might be able to remove some of the reported expectation bias [2], needs to be investigated further.

Generally, an interesting question in the context of interactions between “type” of diagnostic anaesthesia, subjectively judged “efficacy” and patterns of change for specific movement symmetry variables is the following: Are there “biomechanical” differences in relation to a specific “type” of diagnostic anaesthesia, or are observers perceiving or “expecting” specific changes in movement after different diagnostic anaesthesia “types”? “Biomechanical” differences may, for example, result from how diagnostic anaesthesia administered to different structures or regions affects weight-bearing and push-off kinetics. Both force and impulse differences have been documented in association with specific symmetry parameters [11,12], suggesting that quantifying the movement symmetry of the head and pelvis is a valid approach and relates to different patterns in relation to force production.

Expectation bias has been reported for visual assessments of gait changes after diagnostic anaesthesia [2]. The generally small differences between “efficacy” categories, possibly with the exception of some head movements, for example, DHDmin and DHDup after positive low 4-point diagnostic anaesthesia, may contribute to uncertain decisions related to difficulties in perceiving small movement asymmetries [22]. However, it has been argued that additional parameters to the ones investigated here, such as, for example, limb movements, are important for the subjective lameness categorization performed by experts [23]. A recent eye-tracking study, however, indicates a visual focus on the head in a frontal view and on the pelvis in a rear view [24]. Quantitative measurements of limb movements may reveal to what extent these could help with differentiating between diagnostic anaesthesia “efficacy” categories.

The present retrospective study has been conducted at one referral-level hospital with cases assessed by four board-certified veterinary surgeons. The results of this study should hence not be overgeneralized for different patient populations (e.g., age, breed, use) or different facilities (e.g., surface conditions, instrumentation of horses, logistical differences in working on lameness cases). These factors should be included in future large-scale studies.

## 5. Conclusions

The present study has confirmed the “law of sides” by documenting small contralateral changes in pelvic movement symmetry after successful forelimb diagnostic anaesthesia and small ipsilateral head movement changes after successful hind limb diagnostic anaesthesia. While the effects after forelimb diagnostic anaesthesia suggest that compensation may be restricted to pelvic push-off (DPDmax), compensation after hind limb diagnostic anaesthesia is more wide-reaching and affects weight-bearing (DHDmin) and push-off (DHDup). Comparison of our results to previous studies with horses on the lunge suggest that decisions about diagnostic anaesthesia “efficacy” need to take into account differences in weight-bearing and push-off between straight-line and lunge exercises.

Our results have also documented that there are changes in relation to the interaction between the “efficacy” and “type” of diagnostic anaesthesia. These two-way interactions are more prominent after forelimb diagnostic anaesthesia and less so after hind limb diagnostic anaesthesia. Some of the documented patterns seem to suggest that specific “types” of diagnostic anaesthesia are associated with larger (or smaller) movement symmetry changes. Further studies need to clarify to what extent this may be related to veterinary experts learning to recognize specific movement patterns. Alternatively, the “type”-specific movement symmetry changes may be related to “biomechanical” effects in association with specific anatomical structures or regions and their role in providing forces during weight support or during push off. We speculate that including different gaits (e.g., during walk [25]), movement directions (lunge exercise) and surfaces [8,9] may prove useful for creating more distinct movement patterns.

## Figures and Tables

**Figure 1 animals-13-03769-f001:**
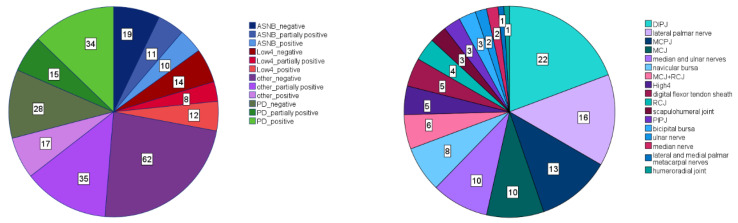
Frequencies of diagnostic anaesthesia “types” in the forelimb data set (total number of forelimb diagnostic anaesthesia instances: N = 265). Left panel: Frequencies (numbers in boxes within each slice of the pie chart) for three most common forelimb “types” (plus category other) for three “efficacy” categories (positive, partially positive and negative). Right panel: Frequency distribution (numbers in boxes within each slice of the pie chart) amongst the “types” summarized in the category “other”. Acronyms: DIPJ: distal interphalangeal joint; MCPJ: metacarpophalangeal joint; MCJ: middle carpal joint; RCJ: antebrachiocarpal joint; High4: high 4-point block; PIPJ: proximal interphalangeal joint.

**Figure 2 animals-13-03769-f002:**
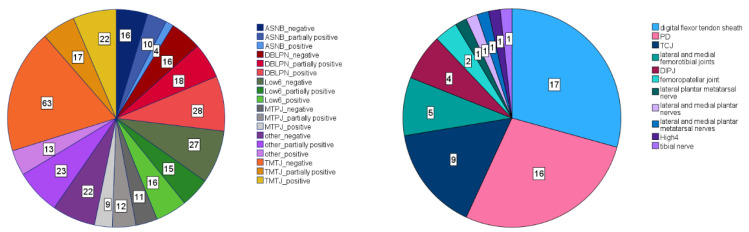
Frequencies of diagnostic anaesthesia “types” in the hind limb data set (total number of hind limb diagnostic anaesthesia instances: N = 342). Left panel: Frequencies (numbers in boxes within each slice of the pie chart) for five most common forelimb “types” (plus category other) for three “efficacy” categories (positive, partially positive and negative). Right panel: Frequency distribution (numbers in boxes within each slice of the pie chart) amongst the “types” summarized in the category “other”. Acronyms: PD: plantar digital; TCJ: tarsocrural joint; DIPJ: distal interphalangeal joint; High4: high 4-point block.

**Figure 3 animals-13-03769-f003:**
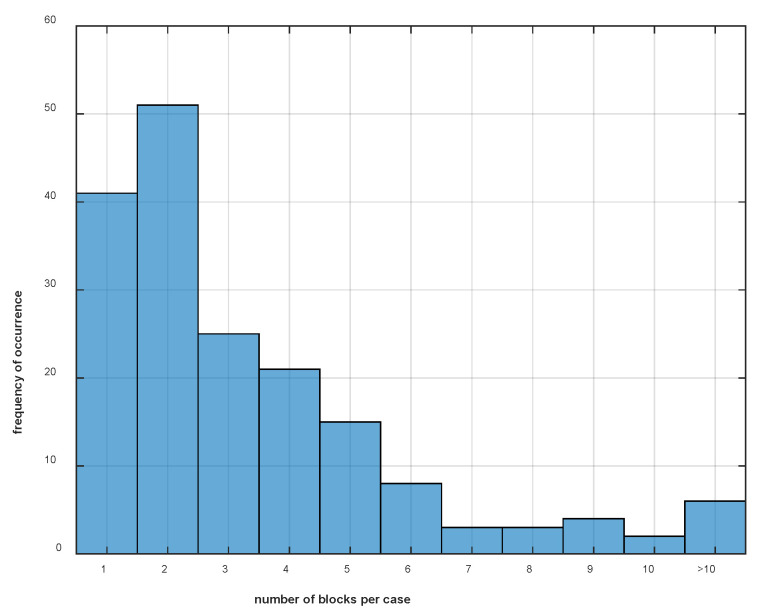
Histogram of the number of diagnostic anaesthesia instances (over one or multiple days) per case for 179 horses included in this study. More than half the cases (92) contributed only one or two instances (diagnostic blocks) to the database; 18 (10%) of the 179 horses contributed seven or more instances to the database.

**Figure 4 animals-13-03769-f004:**
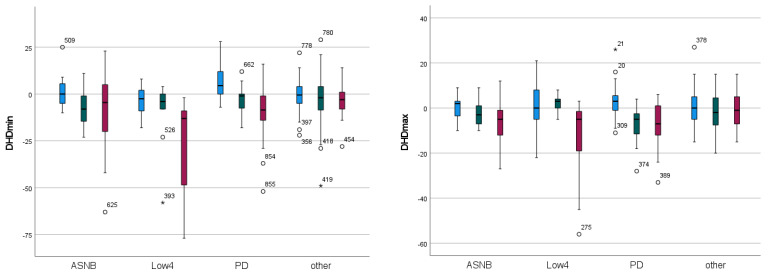
Change in movement symmetry as a function of diagnostic anaesthesia “type” and “efficacy” (blue: ‘negative’; green: ‘partially positive’; red: ‘positive’) administered to **forelimbs** in N = 179 horses quantified with inertial sensor gait analysis. A: DHDmin; B: DHDmax; C: DHDup; D: DPDmax. ◦: values more than 1.5 × interquartile range below the first quartile or above the third quartile with case numbers. *: values more than 3.0 × interquartile range below the first quartile or above the third quartile with case numbers.

**Figure 5 animals-13-03769-f005:**
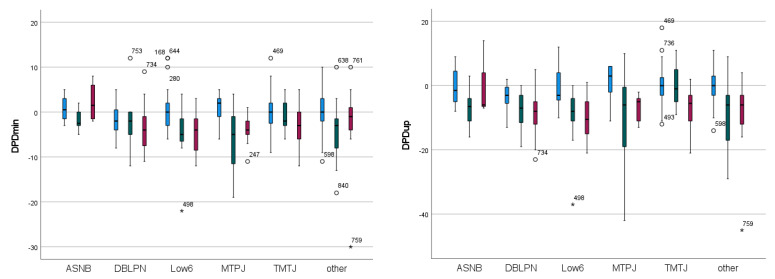
Change in movement symmetry as a function of diagnostic anaesthesia “type” and “efficacy” (blue: ‘negative’; green: ‘partially positive’; red: ‘positive’) administered to **hind limbs** in N = 179 horses quantified with inertial sensor gait analysis. A: DPDmin; B: DPDup. ◦: values more than 1.5 × interquartile range below the first quartile or above the third quartile with case numbers. *: values more than 3.0 × interquartile range below the first quartile or above the third quartile with case numbers.

**Table 1 animals-13-03769-t001:** Results of the mixed model with changes in movement symmetry after **forelimb** diagnostic anaesthesia as a function of “efficacy” and “type” (N = 265 instances of diagnostic anaesthesia from 84 horses). All four parameters indicating significance (DHDmin, DHDmax, DHDup, DPDmax) show a significant two-way interaction between block “type” and “efficacy”. Significant effects (*p* < 0.05) are highlighted in bold face. Acronyms: “Param.”: movement symmetry parameter (see Section 2), “Neg.”: negative, “Part. Pos.”: partially positive, “Pos.”: positive, “ASNB”: abaxial sesamoid nerve block, “Low4”: low 4-point nerve block, “PD”: palmar digital nerve block, “TMTJ”: tarsometatarsal joint block.

Forelimb	Significance			“Efficacy” EMM			“Type” EMM			
Param.	Efficacy	Block Type	2-Way	Neg.	Part. Pos.	Pos.	ASNB	Low4	PD	Other
*DHDmin*	**<0.001**	**<0.001**	**0.023**	1.498	−6.536	−12.160	−5.223	−13.369	−2.272	−2.068
*DHDmax*	**<0.001**	0.121	**0.001**	0.895	−2.355	−7.298	−3.080	−3.627	−4.120	−0.850
*DHDup*	**<0.001**	**0.007**	**0.004**	1.978	−8.430	−18.552	−8.306	−16.077	−6.401	−2.554
*DPDmin*	0.907	0.813	0.603	0.610	0.819	0.885	0.783	1.187	0.630	0.485
*DPDmax*	**0.031**	0.668	**0.003**	−0.554	0.533	1.489	0.245	0.849	0.773	0.090
*DPDup*	0.187	0.601	0.260	0.295	1.474	2.435	1.864	1.860	1.352	0.529
*DHHD*	0.153	0.418	0.560	0.268	−0.443	2.659	−0.492	1.720	1.839	0.245

**Table 2 animals-13-03769-t002:** Results of the mixed model with changes in movement symmetry after **hind limb** diagnostic anaesthesia as a function of “efficacy” and “type” (N = 342 instances of diagnostic anaesthesia from 131 horses). Two pelvic movement parameters (DPDmin, DPDup) show a significant two-way interaction between “type” and “efficacy”. The remaining two pelvic parameters as well as two head movement parameters (DHDmin and DHDup) only show a significant “efficacy” effect but no “type” effect. Significant effects (*p* < 0.05) are highlighted in bold face. Acronyms: “Param.”: movement symmetry parameter (see Section 2), “Neg.”: negative, “Part. Pos.”: partially positive, “Pos.”: positive, “ASNB”: abaxial sesamoid nerve block, “DBLPN”: deep branch lateral plantar nerve block, “Low6”: low 6-point nerve block, “MTPJ”: metatarsophalangeal joint block, “TMTJ”: tarsometatarsal joint block.

Hind Limb	Significance			Block Efficacy EMM			Block Type EMM					
Param.	Efficacy	Block Type	2-Way	Neg.	Part. Pos.	Pos.	ASNB	DBLPN	Low 6	MTPJ	TMTJ	Other
*DHDmin*	**<0.001**	0.090	0.208	0.750	−4.930	−2.329	−1.804	−2.978	−2.667	−5.253	0.177	−0.492
*DHDmax*	0.093	0.304	0.067	1.149	−0.611	1.209	1.290	−0.729	−0.410	2.108	0.875	0.361
*DHDup*	**0.001**	0.446	0.100	1.776	−5.187	−0.532	−0.263	−2.846	−2.744	−2.723	1.206	−0.515
*DPDmin*	**<0.001**	**0.024**	**0.020**	0.230	−3.360	−2.840	0.537	−2.559	−2.821	−3.326	−1.385	−2.387
*DPDmax*	**<0.001**	**0.034**	0.168	−0.662	−3.742	−4.082	−3.088	−3.335	−3.274	−2.095	−1.461	−3.720
*DPDup*	**<0.001**	**0.003**	**0.019**	−0.421	−7.516	−7.134	−2.466	−6.588	−6.479	−5.498	−2.594	−6.515
*DHHD*	**0.001**	0.442	0.787	0.301	−5.270	−3.296	−2.749	−4.687	−2.317	−3.818	−0.868	−2.092

## Data Availability

Data is available on figshare: https://dx.doi.org/10.6084/m9.figshare.6025748.

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
