# Peer review of "Changes in Head and Pelvic Movement Symmetry after Diagnostic Anaesthesia: Interactions between Subjective Judgement Categories and Commonly Applied Blocks"

_animals, 2023, doi:10.3390/ani13243769_

Round 1
Reviewer 1 Report
Comments and Suggestions for Authors
Author Response
REVIEWER 1:
The aim of the present study was to investigate the interaction between the functional effects of diagnostic anesthesia on movement symmetry and the subjective expert judgement ofthe ‘efficacy’ of the administered anesthesia. The strengths of this paper are it compares clinical expertise of the attending veterinarian to measurements identified by use of a gait analysis system. The manuscript describes characteristic patterns following blocking and demonstrates how expert lameness evaluators learn and utilize these patterns while performing lameness examinations.
The paper further aims to demonstrate the effect of diagnostic nerve blocks to desensitize selected areas on limbs of lame horses and combine this information with measurement of changes in movement (of the head and pelvis) using inertial sensor devices to establish different gait patterns. These gait patterns are useful to help examining veterinarians interpret the effect of these blocks on the change in clinical signs. This was a retrospective study which collected data from the medical records of horses presented for either thoracic or pelvic limb lameness. The study analyzed interactions between perceived changes and measured gait parameters on horses trotted in hand. Diagnostic blocking of a suspected site of lameness in a forelimb reduces the asymmetry of head movements (reduced nodding), and improves the horse’s ability to utilize its opposite side hindlimb for push off. Desensitizing a suspected lameness site (source of pain) in a hind limb improves weight bearing and push off strength. This results in reduced hip hike and head nod.
Thank you for the nice summary.
The strengths of the manuscript are demonstration of how use of inertial sensor devices can help localize a lameness and show how the information from that type of examination can be compared to an experienced diagnostician.
Experienced lameness diagnosticians will have made up their mind about the impact of a block well ahead of the time required for a gait analysis system to provide measurements. The present study has confirmed the ‘law of sides’ by documenting contralateral changes in pelvic movement symmetry after successful forelimb diagnostic anesthesia and ipsilateral head movement changes after successful hind limb diagnostic anesthesia. Effects after forelimb diagnostic anaesthesia suggest that compensation may be restricted to pelvic pushoff (DPDmax), compensation after hind limb diagnostic anaesthesia is more wide-reaching and affects weight-bearing (DHDmin) and pushoff (DHDup).
Thank you. We have provided further clarification in the materials and methods section which now explicitly mentions the ‘meaning’ of positive and negative changes in the measured movement symmetry values of head and pelvis in the context of the expected compensatory changes in accordance with the ‘law of sides’.
In my opinion, the primary weakness of the manuscript is that the authors have not clearly defined the importance and usefulness of the ‘law of sides’. Thus, if the authors spent more time on defining the law of sides and describing how it is used by diagnosticians when evaluating a lame horse, it would strengthen the manuscript.
Thank you for highlighting this shortcoming of our manuscript. We have further clarified in the introduction as follows:
“This information can be useful during the clinical lameness examination, for example in horses with signs of both forelimb and hind limb lameness, for the decision making about which limb is a more likely contributor to the displayed lameness pattern. Consequently, when after diagnostic signs of both forelimb and hind limb lameness are disappearing, the horse indeed had shown signs of a compensatory lameness in accordance with the ‘law of sides’.”
For this reviewer, despite reading the manuscript four times, I was unclear on how the article confirms the ‘law of sides’ as stated in the authors conclusions. I was impressed by the statement that experienced diagnosticians can quickly (lees than 30 seconds) make a determination if a block did or did not help and that over time the examiner establishes a mental picture or what a specific lameness looks like before and after use of local anesthesia.
Thank you again for highlighting this weakness of our manuscript. We apologize for having omitted important clarifications in the materials and methods section. We have now further clarified in the materials and methods section the ‘meaning’ of positive and negative changes of movement symmetry of head and pelvis in the context of compensatory movements and the ‘law of sides’:
“As a result of this processing, it is straight forward to draw conclusions about the existence of compensatory changes after successful diagnostic anaesthesia based on the signs of the changes in movement symmetry for head and pelvic movement. After a successful block to a hind limb, negative values are expected for changes in pelvic movement symmetry in accordance with an improvement in hind limb lameness (or a shift of the lameness to the contralateral hind limb). In accordance with the ‘law of sides’ one would also expect negative values for changes in head movement symmetry reflecting an ipsilateral change. In contrast, after a successful block to a forelimb, negative values are expected for head movement symmetry in accordance with an improvement in forelimb lameness (or a shift of the lameness to the contralateral fore-limb). However, now compensatory changes would be expected to cause contralateral changes in pelvic movement symmetry, which according to our data processing would be positive in value.”
Reviewer 2 Report
Comments and Suggestions for Authors
Thank you for embarking on this interesting study. I think the research is sound but that the paper would be strengthened by incorporating more discussion on the clinical application of the results.
Specific comments are as follows:
Title and overall design: The title does not accurately reflect what is presented and to me implies that we are comparing objective and subjective data when it seems that what was done was to evaluate changes in horse movement as measured by inertial sensors. The subjective evaluation was only used to divide the data points into 3 categories of negative, partial and positive response. It does not appear that the objective results were used to corroborate the subjective findings and I wonder why not? If there is objective data, why was the subjective data needed to divide the blocking results into the various categories? It is unclear what happened if the subjective did not align with the objective analysis and this needs to be included in my opinion.
Line 26. Opening sentence reads a bit awkward- consider revising
Line 107- what was used as baseline if sequential blocks were used as sperate data points? This should be described. For example, was an abaxial compared to degree of lameness after a PD or was it compared to baseline?
Line 155 and 159 - The titles above the keys in Figure 1 and 2 should be relabeled as BlockMain and BlockEdited are confusing.
Line 165: Data analysis. It seems the degree of baseline lameness could influence the results. Was this recorded and/or reported anywhere?
Line 183: Is there a way to also diagram/ demonstrate how many of these instances occurred on a single limb versus on multiple limbs?
Line 246: May have been how the table transferred but please correct formatting issues (IE acronym titles span two lines instead of all in one line). May also be easier to read if there were light divisions between the different headings such that Significance is separated a little from Block Efficacy from Block Type.
Line 274: I don’t mind this paragraph but it feels a bit like an opening/ intro statement more than a discussion of the results.
Line 276: The word between seems unnecessary
Line 283: Mention of expertly assigned categories. Would be good to describe in the Materials/Methods sections the experts that were doing the evaluations; IE give reader an indication of their experience level.
Line 295: Expand on this sentence- decrease of 66-78% in pushoff? Before diagnostic analgesia? Just needs minor clarification.
Line 345- 351: This reads as a statement of results and should include insight/ discussion of what it means.
Discussion: Much of the discussion appears to be stating results without sufficient exploration of the what the data means and/or how the authors would interpret the findings. Additionally, the paper would be strengthened by reiterating in the discussion the clinical relevance to the lameness practitioner of the findings.
Author Response
Reviewer 2
Thank you for embarking on this interesting study. I think the research is sound but that the paper would be strengthened by incorporating more discussion on the clinical application of the results.
Thank you for your comments and please see responses to each specific comment below.
Specific comments are as follows:
Title and overall design: The title does not accurately reflect what is presented and to me implies that we are comparing objective and subjective data when it seems that what was done was to evaluate changes in horse movement as measured by inertial sensors. The subjective evaluation was only used to divide the data points into 3 categories of negative, partial and positive response. It does not appear that the objective results were used to corroborate the subjective findings and I wonder why not? If there is objective data, why was the subjective data needed to divide the blocking results into the various categories? It is unclear what happened if the subjective did not align with the objective analysis and this needs to be included in my opinion.
Thank you for highlighting this. Indeed the title of the manuscript can be misinterpreted and most likely will be by uninitiated readers. We have adapted the title as follows:
“Changes in head and pelvic movement symmetry after diagnostic anaesthesia: interactions between subjective judgement categories and commonly applied blocks.”
We believe that this title now reflects the analysis conducted where a retrospective analysis of two-way interactions between the ‘types’ of blocks and the block ‘efficacy’ categories has been conducted.
With respect to the ‘subjective’ versus ‘objective’ data. It is important to keep in mind that this was a retrospective study conducted entirely based on clinical case records. The aim is to objectively analyse the patterns (in head and pelvic movement symmetry) in relation to the ‘overall impression’ of the clinicians about any improvements (or lack thereof) that they noted. This is a ‘pattern recognition’ approach based on the comparison of the ‘subjective impression’ that the clinicians had and have expressed as ‘negative’, partially positive’ and ‘positive’ effects in the cases records and the ‘quantitative gait changes’ measured. We have further clarified in the introduction as follows:
The aim of the present study was to conduct a data-driven retrospective study based on case records investigating the interaction between the functional effects of diagnostic anaesthesia on movement symmetry – quantified by changes in head and pelvic movement symmetry – and the subjective expert judgement of the ‘efficacy’ of the administered anaesthesia, i.e. the overall impression of any improvements noted by the clinicians at the time of the assessment.
Line 26. Opening sentence reads a bit awkward- consider revising
We are struggling to find a better phrasing while staying within the word limits for the abstract.
Line 107- what was used as baseline if sequential blocks were used as sperate data points? This should be described. For example, was an abaxial compared to degree of lameness after a PD or was it compared to baseline?
Thank you. We have added the following sentence to clarify:
“When subsequent blocks were administered to a horse, the values obtained after a specific block were compared to the values obtained before that block rather than to the ‘baseline’ before any blocks had been administered.”
Line 155 and 159 - The titles above the keys in Figure 1 and 2 should be relabeled as BlockMain and BlockEdited are confusing.
We have removed these ‘titles’.
Line 165: Data analysis. It seems the degree of baseline lameness could influence the results. Was this recorded and/or reported anywhere?
Very good point. With the current sample size (which we have provided more detail about in response to reviewer 3) we feel that introducing additional variables into the analysis would not be justified. We have added this point to the discussion. In our current analysis this would have resulted in a reduction of the data set (sample size) since this would have required both the lameness score as well as the ‘percentage improvement’ data and the gait analysis results. Lameness scores and percentage improvement values were not always present simultaneously for all blocks.
We have added the following sentence to the paragraph discussing the ‘magnitude of changes’
“Another factor that has likely influenced the magnitude of changes is the initial lameness grade for the first administered block or the residual lameness grade after subsequent blocks. Including the lameness grade – at each stage of the lameness examination – into the statistical analysis for example as a covariate would appear to be useful to investigate this aspect.”
Line 183: Is there a way to also diagram/ demonstrate how many of these instances occurred on a single limb versus on multiple limbs?
Excellent idea. We have added additional numbers about the number of cases contributing specific sets of data.
“Of the 179 cases, 48 contributed data to forelimb data set only, 95 contributed data to the hind limb data set only and 36 cases contributed data to both forelimb and hind limb data set. Forty-one cases contributed data of one forelimb, 84 cases of one hind limb, 8 cases of both forelimbs, 11 cases of both hind limbs, 13 cases of diagonal pairs of limbs, 14 cases of ipsilateral pairs and 8 cases contributed data of more than 2 limbs.”
Line 246: May have been how the table transferred but please correct formatting issues (IE acronym titles span two lines instead of all in one line). May also be easier to read if there were light divisions between the different headings such that Significance is separated a little from Block Efficacy from Block Type.
Apologies. We completely agree. The Journal has reformatted our original tables. Our original tables were spanning the entire width of the page and had clear delineations between the ‘efficacy’ and ‘type’ results. We will (if the manuscript is accepted) endeavour to revert these changes.
Line 274: I don’t mind this paragraph but it feels a bit like an opening/ intro statement more than a discussion of the results.
Agreed. It is a bit of a repetition. We would love to keep this paragraph as a way of introducing the reader to the discussion section.
Line 276: The word between seems unnecessary
Removed, thank you.
Line 283: Mention of expertly assigned categories. Would be good to describe in the Materials/Methods sections the experts that were doing the evaluations; IE give reader an indication of their experience level.
Thank you for highlighting this omission. We have adapted the manuscript to provide the following information:
“Each case contributing to the database had been assessed throughout the respective lamenesss examination by one of four board-certified veterinary surgeons (Dip. ECVS (and ACVS or ECVSMR)).”
Line 295: Expand on this sentence- decrease of 66-78% in pushoff? Before diagnostic analgesia? Just needs minor clarification.
Thank you. We have further added that these were changes after diagnostic anaesthesia and for different groups of horses.
Line 345- 351: This reads as a statement of results and should include insight/ discussion of what it means.
The line numbers have possibly changed through edits implemented by the Journal? We are not entirely sure what this comment is about. From our submitted version of the manuscript we guess that this comment is about the paragraph describing the ‘positive’ changes after forelimb PD blocks? We are not entirely sure what we could add here over and above the statement that PD block that has been judged as negative the horses appear to show signs of a ‘worsening’ lameness and that this is of small magnitude.
Discussion: Much of the discussion appears to be stating results without sufficient exploration of the what the data means and/or how the authors would interpret the findings. Additionally, the paper would be strengthened by reiterating in the discussion the clinical relevance to the lameness practitioner of the findings.
In our original manuscript, we had attempted to put our results into the context of previously published studies in the area of diagnostic anaesthesia and this has required discussing the results of our data-driven study. In our opinion this goes beyond the mere ‘stating of results’ and we do not want to over-interpret or over-generalize the results of our study. We have, in response to the reviewer's comment, added a couple of sentences to clarify the relevance of our results for interpreting the quantified gait changes after diagnostic anaesthesia in particular in the context of the ‘law of sides’ and which changes in quantitative parameters may be expected based on our findings. We hope this is satisfactory.
Reviewer 3 Report
Comments and Suggestions for Authors
While understanding of diagnostic methods for effectively detecting lameness within the horse is a vital part of the veterinary industry and the approach to this topic is unique in nature, the manuscript falls short in presenting information in a manner that can allow for replication and further understanding of application and determination of sound conclusions. The title is unclear as "differences in subjective judgement" sounds as if this study will be applying various approaches of subjective judgement methodology rather than comparing the one approach to the use of sensor units. Further, it would be helpful to be specific as to what is meant by "subjective judgement". The "subjective judgement" is the use of visual assessment method using a standardized categorization method on determining level of lameness. This needs to be clarified within the title and it would be helpful to clarify this more within the summary, abstract, and introduction as to the specific methodology utilized since it is unique from other lameness categorization/scoring methods. Also, within the summary and abstract, even within the results, it needs to be clear what the sample population is for each aspect of this study. Specifically, numbers utilized for forelimb assessments versus that of hindlimb assessments along with specific numbers for each block utilized within the study. These blocks need to be listed within the abstract. Starting from the summary and abstract, clearly define what it means to be labeled as an "expert". Further, within the methods, include the number of veterinarians utilized for this study and what the inclusion/exclusion criteria was for these individuals to be classified as "experts".
As for the introduction, further information, as mentioned above, must be given on the standardized visual assessment and lameness categorizing method utilized for this study and how that compares to other similar methods utilized in other countries. Would a scoring method be more effective than the method utilized for this study? Lay out all the options of lameness analysis available for the veterinarian including not only visual methods, but also methods utilizing technology such as kinematic and kinetic analysis. What are the pros and cons of these various methods and what has research found concerning limitations? Researchers utilized inertial measurement units to determine reliability of visual assessment methods, but are these units 100% reliable? Further background information is needed within the introduction addressing these questions.
The methods section falls short as to what was actually done with these horses concerning the lameness assessment methods. A seven year study period opens up the potential of variability between how the data was collected. Clearly define how the visual assessment method was performed. What was the standardized methodology applied concerning how the animal was tracked and what was observed for the categorization performed. What was the order in how assessment was done? What was the experience of these veterinarians and of the individuals that assisted with tracking these animals during the visual assessment. What type of training was required and was it standardized? What was the experience and training of those that applied the inertial measurement units? What standardized methodology was applied to ensure consistency in unit application between each animal? In the seven years, were there updates in the unit software and even the units themselves that could potentially impact unit application, data collection, and data output? What were the horses used for this study? A table within supplementary material giving further information on the horses and their diagnosed condition that resulted in the lameness should be given. This should include details concerning the length of this lameness issue as the type of condition and the length of this condition can cause adaptations to gait that will not resolve completely with diagnostic anesthesia. Were cases such as this excluded from the study? Geriatric cases will have compounding factors that will not resolve with diagnostic anesthesia along with neurological conditions, and as such, were these factors a consideration? Were bilateral lameness included? Further information on the animals utilized including specific numbers for each aspect of this study will help to identify if additional compounding factors may have influenced reliability of the data. Further descriptions of the blocks applied within this study are necessary including the methodology utilized for the blocks and assurances as to the qualifications of the individuals performing the blocks so that consistency can be verified between the subjects utilized for the study. Was velocity maintained consistent between the visual assessment and assessment utilized with the units including all assessments utilized with the blocks as velocity can impact gait performance and accentuate lameness characteristics? Figures 1 and 2 are too small to read and need to be further clarified within the text.
As for the results, the specific sample population and associated numbers needs to be given in a clear manner for each step in the analysis process and this includes within the tables and figures. Seems that there were 179 horses for the fore and hind analysis, but does this mean that the same 179 horses were used for both, and if so, does that mean that these horses had both fore and hind lameness? This information is unclear. Table titles should not include the results, thus, shorten titles and include that information within the text. Figure 3 should be included within the information on each horse to clarify as to whether the horse had fore or hind limb lameness and associated blocks. Unclear as to what the figure is referring to concerning "frequency of occurrence". Not sure what value this figure holds, although it appears that some horses (18) were used multiple times for this study, and if so, one might question the use of a horse multiple times within a study as that would seem that the data was skewed concerning the type of lameness being treated and the veterinarians and staff treating these specific animals. The actual number of horses for forelimb analysis and those for hindlimb analysis without duplicates needs to be given. Duplicates needs to be removed.
As for the discussion, without further clarification within the methodology, it is hard to determine what conclusions should be drawn. Limitations may need to incorporate some the above concerns that were pointed out within this review. Further, the use of a single veterinary hospital for data collection can introduce potential common problems within the facility such as shortcomings in the expertise, technology, and/or facilities and/or introduce limits within the clientele being seen. As such, this needs to be addressed. Finally, the conclusions need to focus only on what exact conclusions can be made concerning the results given so that speculation isn't drawn. Suggestion of further studies should be given within the discussion section with research supporting the need for further work in these areas.
Comments on the Quality of English LanguageSee comments above. Overall, clarification is needed throughout. Further, reduction of length of sentences will assist in some areas as run-on sentence structure is seen throughout.
Author Response
Reviewer 3:
While understanding of diagnostic methods for effectively detecting lameness within the horse is a vital part of the veterinary industry and the approach to this topic is unique in nature, the manuscript falls short in presenting information in a manner that can allow for replication and further understanding of application and determination of sound conclusions. The title is unclear as "differences in subjective judgement" sounds as if this study will be applying various approaches of subjective judgement methodology rather than comparing the one approach to the use of sensor units.
Thank you. We have changed the title of the manuscript since in hindsight we agree that the title is open for misinterpretation.
Further, it would be helpful to be specific as to what is meant by "subjective judgement". The "subjective judgement" is the use of visual assessment method using a standardized categorization method on determining level of lameness. This needs to be clarified within the title and it would be helpful to clarify this more within the summary, abstract, and introduction as to the specific methodology utilized since it is unique from other lameness categorization/scoring methods.
We disagree with the reviewer here. The categorization into three ‘subjective categories’ has been utilized before and we have clearly stated this in the introduction:
Observers may utilize percentage values to express their judgement about the ‘efficacy’ of diagnostic anaesthesia for altering movement and a mapping of 0 to 30%, >30 to 70% and >70% improvement has been suggested for the three categories [5]. This refers to a published manuscript (Hawkins, A.; O’Leary, L.; Bolt, D.; Fiske‐Jackson, A.; Berner, D.; Smith, R. Retrospective Analysis of Oblique and Straight Distal Sesamoidean Ligament Desmitis in 52 Horses. Equine Vet J 2021, doi:https://10.1111/evj.13438.) and the same categorization has been used for analysis of diagnostic anaesthesia data previously in two manuscripts:
1) Marunova, E., Hoenecke, K., Fiske-Jackson, A., Smith, R.K.W., Bolt, D.M., Perrier, M., Gerdes, C., Hernlund, E., Rhodin, M., Pfau, T., 2022. Changes in Head, Withers, and Pelvis Movement Asymmetry in Lame Horses as a Function of Diagnostic Anesthesia Outcome, Surface and Direction. Journal of Equine Veterinary Science 118, 104136. https://doi.org/10.1016/j.jevs.2022.104136.
2) Pfau, T., Bolt, D.M., Fiske-Jackson, A., Gerdes, C., Hoenecke, K., Lynch, L., Perrier, M., Smith, R.K.W., 2022. Linear Discriminant Analysis for Investigating Differences in Upper Body Movement Symmetry in Horses before/after Diagnostic Analgesia in Relation to Expert Judgement. Animals 12, 762. https://doi.org/10.3390/ani12060762
Also, within the summary and abstract, even within the results, it needs to be clear what the sample population is for each aspect of this study. Specifically, numbers utilized for forelimb assessments versus that of hindlimb assessments along with specific numbers for each block utilized within the study.
Information about the number of forelimb and hind limb blocks and the number of blocks of each ‘type’ had been provided in the original manuscript as follows:
- numbers of negative, partially positive and positive blocks for the main categories separately for forelimb and hind limb blocks (Figures 1+2). We apologize if these were too small to be read. We have increased the font size and added some additional information in the main text.
- numbers of specific blocks of the ‘other’ category for forelimb ad hid limb blocks (Figures 1+2).
- numbers of instances of diagnostic anaesthesia (607 total, 265 to forelimbs, 342 to hind limbs, first paragraph of the results section.
- Distribution of the numbers of blocks per case (Figure 3).
We have added additional information in the new manuscript:
“Of the 179 cases, 48 contributed data to the forelimb data set only, 95 contributed data to the hind limb data set only and 36 cases contributed data to both forelimb and hind limb data sets. Forty-one cases contributed data of one forelimb, 84 cases of one hind limb, 8 cases of both forelimbs, 11 cases of both hind limbs, 13 cases of diagonal pairs of limbs, 14 cases of ipsilateral pairs and 8 cases contributed data of more than 2 limbs.”
We have included the specific blocks and the forelimb and hind limb sample size into the abstract now. Thank you for highlighting this omission.
These blocks need to be listed within the abstract. Starting from the summary and abstract, clearly define what it means to be labeled as an "expert". Further, within the methods, include the number of veterinarians utilized for this study and what the inclusion/exclusion criteria was for these individuals to be classified as "experts".
Thank you. We have provided additional information about the qualifications of the four experts used:
“Each case contributing to the database had been assessed throughout the respective lameness examination by one of four board-certified veterinary surgeons (Dip. ECVS (and ACVS or ECVSMR)).”
As for the introduction, further information, as mentioned above, must be given on the standardized visual assessment and lameness categorizing method utilized for this study and how that compares to other similar methods utilized in other countries. Would a scoring method be more effective than the method utilized for this study? Lay out all the options of lameness analysis available for the veterinarian including not only visual methods, but also methods utilizing technology such as kinematic and kinetic analysis. What are the pros and cons of these various methods and what has research found concerning limitations? Researchers utilized inertial measurement units to determine reliability of visual assessment methods, but are these units 100% reliable? Further background information is needed within the introduction addressing these questions.
We feel that the suggestions of the reviewer, in particular the inclusion of advantages/disadvantages of different methods are not within the realm of this manuscript. There are a number of comparison/validation studies that have been published on the use of IMUs suitable for clinical lameness examinations in horses and the use if IMUs (and more and more also optical methods) is now an accepted standard for quantifying gait changes in relation to weight bearing and pushoff asymmetry in clinical practice. We are not indicating that the IMU based method is superior to the visual assessment but we are aiming to employ a data-driven approach to further analyze whether there are any ‘patterns’ in the data that may provide further clues about when veterinarians consider diagnostic anaesthesia as successful.
The method used for categorization of the data into three categories has been published and has been employed in two studies previously (see above).
The methods section falls short as to what was actually done with these horses concerning the lameness assessment methods. A seven year study period opens up the potential of variability between how the data was collected. Clearly define how the visual assessment method was performed. What was the standardized methodology applied concerning how the animal was tracked and what was observed for the categorization performed. What was the order in how assessment was done?
This was not a prospective study in which all of these factors can be specified and controlled. The point of this study was to retrospectively investigate whether there are any associations between how the veterinary surgeons perceive any improvements in gait, how there is a possible influence on this perception by the different anatomical structures/regions blocked and how this may or may not influence the mechanics of movement as quantified through changes in head and pelvic movement.
What was the experience of these veterinarians and of the individuals that assisted with tracking these animals during the visual assessment. What type of training was required and was it standardized? What was the experience and training of those that applied the inertial measurement units?
We have included information about the qualifications of the veterinary surgeons. We have also clarified that sensors had been attached by trained personnel.
What standardized methodology was applied to ensure consistency in unit application between each animal? In the seven years, were there updates in the unit software and even the units themselves that could potentially impact unit application, data collection, and data output? What were the horses used for this study? A table within supplementary material giving further information on the horses and their diagnosed condition that resulted in the lameness should be given. This should include details concerning the length of this lameness issue as the type of condition and the length of this condition can cause adaptations to gait that will not resolve completely with diagnostic anesthesia. Were cases such as this excluded from the study?
We agree that additional information about the horses is ultimately of use. This initial study however was designed to implement a ‘simple’ data driven approach to identify whether there are (across all case data that were accessed retrospectively) any effects that associate perceived improvement of lameness, anatomical region/structure and quantitatively measured gait changes. It was not the aim of the study to further analyze any additional factors, which will require a much larger data base that then can be used to ‘model’ many more factors. Similarly we have restricted the analysis to diagnostic anaesthesia of the ‘main’ blocks utilized here for which there were 30 or more instances in the database. We completely agree that ultimately all the factors mentioned should be ‘modeled’, here we restrict our analysis to the interaction between the ‘main blocks’ and their ‘subjectively perceived’ effect.
Geriatric cases will have compounding factors that will not resolve with diagnostic anesthesia along with neurological conditions, and as such, were these factors a consideration? Were bilateral lameness included?
See above. We agree that more factors need to be considered ultimately. Yes, bilateral lameness was included. We have further clarified the ‘proportion’ of data that were related to horses that contributed data from a single blocked limb, from contra-lateral/diagonal and ipsilateral pairs of blocked limbs. All data sets compared the quantitative data gathered before the block to the data gathered after the block. This has been clarified in the materials and methods section.
Further information on the animals utilized including specific numbers for each aspect of this study will help to identify if additional compounding factors may have influenced reliability of the data. Further descriptions of the blocks applied within this study are necessary including the methodology utilized for the blocks and assurances as to the qualifications of the individuals performing the blocks so that consistency can be verified between the subjects utilized for the study.
We have provided information about the qualifications of the veterinary surgeons in charge of the lameness workups (all were board certified: all ECVS, some additionally ACVS or ECVSMR).
Was velocity maintained consistent between the visual assessment and assessment utilized with the units including all assessments utilized with the blocks as velocity can impact gait performance and accentuate lameness characteristics?
This is a retrospective study of case data not a prospective study in which all conditions can be defined a-priori. It is not within the remit of the current study to further analyze specific confounding factors. The aim of this study is a data-driven analysis of the interaction between what veterinary surgeons consider a successful or less successful diagnostic block, what anatomical structure/region was blocked and how this may or may not be associated with specific changes in weight bearing and pushoff (and associated compensatory changes).
Figures 1 and 2 are too small to read and need to be further clarified within the text.
Apologies, we have increased the font size of Figures 1 and 2 and have added some further description to the text mentioning the numbers of diagnostic anaesthesia instances for the main forelimb and hind limb blocks.
As for the results, the specific sample population and associated numbers needs to be given in a clear manner for each step in the analysis process and this includes within the tables and figures. Seems that there were 179 horses for the fore and hind analysis, but does this mean that the same 179 horses were used for both, and if so, does that mean that these horses had both fore and hind lameness? This information is unclear.
The information about the sample sizes for forelimb ad hind limb diagnostic anaesthesia that had been provided in the original manuscript (first paragraph of results section) has been augmented.
We have added the number of horses contributing data to the forelimb and to the hind limb data set and are providing sample sizes (instances of diagnostic anaesthesia) and number of horses contributing data in the information accompanying table 1 and table 2.
Table titles should not include the results, thus, shorten titles and include that information within the text. Figure 3 should be included within the information on each horse to clarify as to whether the horse had fore or hind limb lameness and associated blocks. Unclear as to what the figure is referring to concerning "frequency of occurrence". Not sure what value this figure holds, although it appears that some horses (18) were used multiple times for this study, and if so, one might question the use of a horse multiple times within a study as that would seem that the data was skewed concerning the type of lameness being treated and the veterinarians and staff treating these specific animals. The actual number of horses for forelimb analysis and those for hindlimb analysis without duplicates needs to be given. Duplicates needs to be removed.
We have provided additional information about how many horses contributed data sets of individual (forelimbs or hid limbs), of pairs of limbs (contra-lateral, diagonal, ipsilateral) or of more than two limbs. Each instance of diagnostic anaesthesia that has been administered as part of the clinical lameness examination is a separate data entry. The use of one horse’s data for more than one data entry simply reflects the nature of the lameness examination in which subsequent blocks are administered. We have clarified in the materials and methods section that our data sets all consist of measurements before the block and measurements after the block.
Figure 3 states that it is displaying a “Histogram of the number of diagnostic anaesthesia instances per case for 179 horses included in this study”. The ‘frequency of occurrence’ (a standard term used to define the y-axis in a histogram, for example as opposed to ‘percentage of occurrence’) refers to the actual number of diagnostic anaesthesia instances and we have clarified that these instances can occur over one or multiple days.
As for the discussion, without further clarification within the methodology, it is hard to determine what conclusions should be drawn. Limitations may need to incorporate some the above concerns that were pointed out within this review. Further, the use of a single veterinary hospital for data collection can introduce potential common problems within the facility such as shortcomings in the expertise, technology, and/or facilities and/or introduce limits within the clientele being seen. As such, this needs to be addressed.
We have added a short paragraph at the end of the discussion.
Finally, the conclusions need to focus only on what exact conclusions can be made concerning the results given so that speculation isn't drawn. Suggestion of further studies should be given within the discussion section with research supporting the need for further work in these areas.
We would be very grateful for more specifically pointing out where the reviewer is referring to ‘speculations’. In our opinion, we are very careful with not overinterpreting our results. We have, in response to the other reviewers, introduced some more detailed explanations about the relevance of our results in the context of the ‘law of sides’.
Comments on the Quality of English Language
See comments above. Overall, clarification is needed throughout. Further, reduction of length of sentences will assist in some areas as run-on sentence structure is seen throughout.
Please provide us with specific examples of sentences that need improvement. The remaining reviewers have not highlighted any issues with the quality of the English language.
Round 2
Reviewer 3 Report
Comments and Suggestions for Authors
Authors are commended on the work they have put forth within the revisions. Nevertheless, much of the suggestions given within the initial review were not addressed within the revisions, but instead, were addressed within the comments given back to the reviewer. Some of these comments would be of value if introduced within the manuscript itself. Nonetheless, the significant limitations within the previous version of this manuscript were the methodology and the title and both of these were addressed to the point that the manuscript meets the minimal standards for moving forward with publication. The work within this manuscript holds value, but readers will have a difficult time appreciating the implication of the work due to issues within how the work was presented, specifically within the introduction and the discussion sections including the conclusions. Refer to the previous comments of the reviewer for further addressing these sections. Some of these reservations were also pointed out by the other reviewers. In any case, the presentation has some to do with the amount of information trying to be given within the manuscript and the limitations associated with where this data was acquired. Trying to do a retrospective study is difficult, and as such, the authors are commended for what they were able to do with the data that was available. Further, the amount of information that was available over a multi-year study can make it hard as to what to focus on as a 'take-home' message, thus, again, the authors should look specifically at the discussion to determine what results are being presented in a clear, organized fashion to ensure the areas of the work aren't being overlooked by the manner of the presentation.
Comments on the Quality of English LanguageSee comments above.
Author Response
Thank you very much to reviewer 3 for the additional comments. We have implemented some further changes with the main aim of providing some conclusions/interpretations with more direct practical relevance in particular with respect to compensatory changes and the differences between the current study (straight-line) and previous studies with lunge exercise.
Comment: Authors are commended on the work they have put forth within the revisions. Nevertheless, much of the suggestions given within the initial review were not addressed within the revisions, but instead, were addressed within the comments given back to the reviewer. Some of these comments would be of value if introduced within the manuscript itself. Nonetheless, the significant limitations within the previous version of this manuscript were the methodology and the title and both of these were addressed to the point that the manuscript meets the minimal standards for moving forward with publication.
Response: Thank you for your comments. We recognize that it was not possible to act on every one of the original comments of this reviewer and thank the reviewer for recognizing that title and methods have been addressed.
Comment: The work within this manuscript holds value, but readers will have a difficult time appreciating the implication of the work due to issues within how the work was presented, specifically within the introduction and the discussion sections including the conclusions. Refer to the previous comments of the reviewer for further addressing these sections.
Response: Thank you for your comment. We had added a sub-section to the introduction clarifying a little more how the law of sides may be useful (see here):
“This information can be useful during the clinical lameness examination, for example in horses with signs of both forelimb and hind limb lameness, for the decision making about which limb is a more likely contributor to the displayed lameness pattern. Consequently, when after diagnostic signs of both forelimb and hind limb lameness are disappearing, the horse indeed had shown signs of a compensatory lameness in accordance with the ‘law of sides’.”
We have now added more details to the introduction of the revised version:
“In this context, it appears interesting to further investigate to what extent a more de-tailed analysis of quantitative gait collected during clinical lameness examinations may provide further scope for drawing specific conclusions. For example, does a desensitization of different anatomical areas or structures result in different movement patterns of the horse and how is this reflected in the judgement of veterinary experts.”
Comment: Some of these reservations were also pointed out by the other reviewers. In any case, the presentation has some to do with the amount of information trying to be given within the manuscript and the limitations associated with where this data was acquired. Trying to do a retrospective study is difficult, and as such, the authors are commended for what they were able to do with the data that was available. Further, the amount of information that was available over a multi-year study can make it hard as to what to focus on as a 'take-home' message, thus, again, the authors should look specifically at the discussion to determine what results are being presented in a clear, organized fashion to ensure the areas of the work aren't being overlooked by the manner of the presentation.
Response: Thank you again for your considerate comments. We had added more relevance to the ‘law of sides’ in the first revision by referring more explicitly to the ‘law of sides’ for example in the results section as well as in the discussion and had structured the discussion into distinct sections (two sections dealing with compensatory movements, two sections dealing with the ‘type’ of diagnostic anaesthesia and one section with the ‘magnitudes’ of changes. We feel that the structure of the discussion follows a logical theme aiming at discussing distinct aspects about compensatory movements and the differences between specific blocks. We have now, in response to your additional comments, added some further clarifications, in particular in the discussion section, aiming at making our manuscript at least a little more ‘accessible’; we realize that the nature of these results is somewhat 'technical' and 'quantitative'. We have tried to add more specific relevance to the effects that should be ‘expected’ during a clinical lameness examination after diagnostic anaesthesia.
For example in the section on compensation after forelimb blocks):
“As a result, it appears important to adjust the interpretation of quantitative gait data after diagnostic anaesthesia specifically to whether the horse has been assessed on the straight or n the circle. The present study would suggest that a contra-lateral compensatory pushoff effect (DPDmax) can be expected after forelimb diagnostic anaesthesia.”
In the section on compensation after hind limb blocks:
“This again emphasizes the potential of altered ‘mechanics’ during circular movement as a diagnostic aid for veterinarians and, in analogy to forelimb diagnostic anaesthesia, interpretation of quantitative gait data after diagnostic anaesthesia should be specific to the whether the horse is exercised in straight lines or on the lunge.”
and in the conclusion:
“Comparison of our results to previous studies with horses on the lunge suggest that decisions about diagnostic anaesthesia 'efficacy’ need to take into account differences in weight bearing and pushoff between straight-line and lunge exercise.”